# Looking for Solutions to the Pitfalls of Developing Novel Antibacterials in an Economically Challenging System

**Gilles Courtemanche [1]**, **Rohini Wadanamby [2]**, **Amritanjali Kiran [3]**, **Luisa Fernanda Toro-Alzate [4]**, **Mathew Diggle [5]**, **Dipanjan Chakraborty [6]**, **Ariel Blocker [7]** and **Maarten van Dongen [8,*]**

1. BIOASTER, 75015 Paris, France; Gilles.COURTEMANCHE@bioaster.org
2. Lanka Hospital Diagnostics, Colombo 5, Kandy 20000, Sri Lanka; rohiniw@lhd.lk
3. Center for Cellular and Molecular Platforms (C-CAMP), Bengaluru 560065, India; amritanjali.kiran@gmail.com
4. Master of Public Health at Royal Tropical Institute, 1092 AD Amsterdam, The Netherlands; lutoro@gmail.com
5. ProvLab (APL), University of Alberta Hospital Edmonton, Edmonton, AB T6G 2J2, Canada; Mathew.Diggle@albertaprecisionlabs.ca
6. ZeiniX Life Sciences, Bengaluru 562123, India; dipanjan@zeinixlife.com
7. Severe Bacterial Infections Cluster and Bacteriomics Platform, Evotec ID, 69007 Lyon, France; ariel.blocker@evotec.com
8. AMR Insights, 1017 EG Amsterdam, The Netherlands
* Correspondence: maarten@amr-insights.eu

**Abstract:** The increase in antibacterial resistance (ABR) currently equates in the minds of many with the distant fear that certain antibiotics will not work in 30 years on certain bacteria found in places the majority of us never go to. However, in reality, rising ABR already seriously threatens the effectiveness of compounds with which we treat common bacterial infections, which means that ABR is currently and will continue to undermine the foundations of modern medicine, including surgery and cancer treatment in hospitals, cities and countries across the world. That is why ABR is widely considered a global threat and one of the biggest problems of our current civilization. Conversely, antibiotic developments to market are few. Therefore, in this paper, we have illustrated the barriers to antimicrobial R&D the following questions and provided solutions to effective antimicrobial R&D.

**Keywords:** antibacterial; antibiotic; resistance; bacteria; infection; incentive; opinion; market; economic model; AMR; ABR

## 1. Introduction

Antimicrobial resistance (AMR), the natural selection of bacteria, viruses, fungi and parasites, which is able to resist antimicrobial substances normally active against them, is one of the greatest threats of the 21st century. While resistance to antimalarials and antivirals is already a significant issue worldwide [1,2], by far the biggest issue currently is antibacterial resistance (ABR). This is due to a number of contributing factors, including the increased number of bacterial infections we are able to diagnose, how cost-effective most antibiotics are and how extensively they are used in most areas of medicine. In the United States alone, the Centers for Disease Control and Prevention (CDC) estimated that ABR caused an estimated 35,900 deaths in 2019, at least 12,800 of which were linked to a single pathogen, antibiotic-resistant *Clostridium difficile* [3]. Due to significant under reporting from a largely privatised health system, the real figure is likely to be closer to 160,000 [4]. In 2015 by comparison, France, which has one fifth of the U.S. population and a lower incidence of most multidrug-resistant (MDR) bacteria, reported 5543 deaths [5]. In low- and middle-income countries (LMICs), poor hygiene conditions and lack of infection prevention and control (IPC) increase the demands for antibacterial drugs [6]. Unfortunately, the use of broad-spectrum antibiotics in the absence of any diagnostic stewardship (one of the leading causes of ABR) is increasing in LMICs worldwide, reaching figures of close to 100%

in sub-Saharan Africa [7]. So far, however, the available data on ABR-related deaths in LMICs are sparse, due to inadequate surveillance and administration programs.

Even beyond the rising mortality rate, the negative economic impact of ABR highlights the need to act to solve this problem. In the European Union / European Economic Area Countries, the cost of ABR in health systems in 2015 was USD Purchasing Power Parity (PPP) 1.5 billion per year, with an estimated increase in the next 30 years of USD PPP 60 billion. In the United States, Canada and Australia, these expenses will reach USD PPP 74 billion [8]. In LMICs such as Turkey, Thailand and Colombia, the cost of medical care due to ABR can reach more than USD 35,000 per patient in contrast to in countries such as Senegal, where the cost of ABR in health systems is around USD 1000 per patient. However, it is important to consider these apparent differences in the context of the different social and economic characteristics of these countries.

## 2. Five Core Strategies to Combat Rising ABR

The reduction and, where feasible, elimination of ABR requires the implementation of five core strategies to improve the prevention of infectious diseases with the aid of vaccines and better sanitisation and hygiene, such that people do not be become sick as often and do not need antibacterial agents as regularly:

1.  Applying antibacterial stewardship: a better monitoring of the prescription and use of antibacterial drugs in humans, animals and agriculture;
2.  Improving diagnostics to determine the identity and antibiotic susceptibility of bacteria infecting patients such that appropriate antibacterial drugs can be administered right from the start of any treatment; this primarily means combining the extensive expertise within clinical bacteriology by using the development and ubiquitous distribution of point-of-care diagnostics.
3.  Developing novel classes of antibacterial drugs against bacterial species in which significant antibiotic resistances are observed worldwide;
4.  Developing alternative treatments for infectious diseases such as phage therapy [9].

Combating ABR should be based on a holistic, One Health approach in which humans, the animal kingdom, the plant world and the greater environment are considered as interconnected. Apart from the abovementioned core strategies, the uncontrolled release of antibacterial drugs in the environment must be addressed, monitored and ultimately prevented. The real solution to ABR lies in the continuous, effective and balanced global application of all these five core strategies. This demands awareness, understanding and commitment from professionals, governments and the public alike. The magnitude and impact of ABR is not adequately recognized globally, where accountability is therefore largely sporadic. However, as illustrated currently by emerging viral infections such as SARS-CoV2, MDR and pan-drug-resistant (PDR) bacterial pathogens do not recognize any form of border control. Therefore, it is necessary to identify and engage stakeholders across the world, and these should stem from the majority of geographical areas. In addition, global organisations such as the World Health Organization (WHO), the World Organization for Animal Health (OIE), the Food and Agriculture Organisation (FAO), the United Nations Environment Program (UNEP) and other governance systems should further align towards the One Health approach.

## 3. Developing Novel Antibacterial Drugs as One of the Core Strategies to Combat ABR

The discovery of antibiotics is considered one of the most significant milestones in medical research. However, by the very nature of Darwinian selection, the consequence of antibiotic usage, resistance to antibiotics, and antibacterial drugs at large is inevitable. As a consequence, the discovery of novel antibacterial drugs needs to be a continuous process with significant investment, as every new antibacterial will have a finite, largely predictable life span before resistance develops. Historically, the emergence of resistance to clinical antibacterial drugs has been combated with either the modification of existing antibiotic classes in order to limit or reduce cross-resistance to existing drugs or, though

more rarely, the introduction of new compounds and classes. The first approach only provides a short-term solution because: (1) cross-resistance, i.e., resistance to related antibiotics, arises more readily from a preselected gene pool [10,11]; (2) it cannot overcome multiple existing resistance mechanisms and hence cannot control co-resistance, i.e., the growing number of multidrug-resistant pathogens. However, the latter strategy is crucial in combating ABR. Novel antibacterial drugs that show no cross-resistance with or co-resistance to existing classes of antibiotics are consistently required for all disease-causing bacteria and particularly those included in the ESKAPE group (*Enterococcus faecium*, *Staphylococcus aureus*, *Klebsiella pneumoniae*, *Acinetobacter baumannii*, *Pseudomonas aeruginosa* and *Enterobacter* species).

To be truly innovative, novel antibacterial drugs must fulfil at least one of the following parameters and preferably several as following: (a) the absence of cross-resistance to existing antibacterials; (b) new chemical class; (c) new molecular target; (d) new mechanism of action (MoA).

Unfortunately, after the golden age of the discovery of new antimicrobials in the 1940s and 1970s, when several distinct novel classes of antimicrobials largely o riginating from bacterial and fungal natural products were licenced, few new classes have been successfully brought to market (Figure 1). In the 1980s, the total number of antimicrobials approved fell sharply, increasing only slightly between 2011 and 2016.

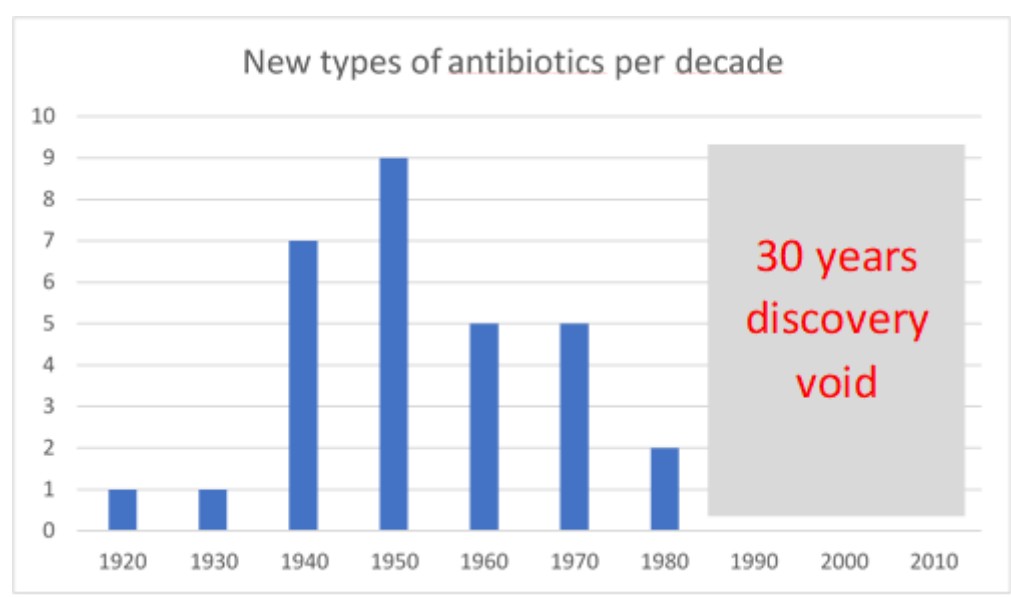

**Figure 1.** New types of antibiotics per decade [12].

## 4. Antimicrobial Pipeline

According to the reports on 30 June 2019 by the PEW Charitable Trusts [13], only 13 antimicrobial candidates were in phase III (Ph3) clinical trials, out of which only 3 were based on novel classes. Meanwhile, in the past five years, no new antimicrobial has been approved, which can address/target one of the WHO critical pathogens. Moreover, approximately 60% of the projects involve nontraditional therapies, of which the development and regulatory pathways therefore remain to be defined and which may not validate existing preclinical models and/or fail in current clinical trial set-ups. Furthermore, an ever increasing number of more modern strategies now focus on the development of pathogen-specific and patient-specific therapies, which as demand develops will require support from novel health-care diagnostic systems.

The most important question, however, remains whether the current, renewed antibacterial pipeline is broad and flexible enough to combat the escalating global threat of ABR. A review of these 407 highly diverse projects in the current preclinical pipeline indicates

that 81% of the involved institutions are small and medium-sized entities (SMEs), of which 60% have less than 10 employees. Therefore, most of the world's important preclinical antibacterial pipelines is supported by SMEs with minimal financial, technical and human resources. Only a limited number of large international pharmaceutical companies still have small numbers of antimicrobial products in the pipeline. The vast and increasing sums of money required to move through various clinical trial phases and beyond commercialisation mean the innovation we so desperately need in this area is getting stifled at both national and international levels.

## 5. Cost of Novelty

There are several reasons why the number of novel antibiotic classes approved for human use has fallen sharply. These can be identified as follows.

### 5.1. At Research Level

5.1.1. New Compound Screenings Are Required

Indeed, costly failures were encountered by large pharmaceutical companies that deployed target-based in vitro high-throughput screening (HTS) approaches in the late 1990s and early 2000s [14]. While over 90% of existing antibiotics are natural products, i.e., derived from fungal or bacterial metabolisms, the companies developing these high-throughput screens, which were initially performed with success for the in vitro screening of compounds on eukaryotic targets and were also chosen to use their libraries of small molecules in bacteria. However, these two chemical spaces are radically different [15]. Therefore, the major issue observed was the bacterial permeability barrier, in particular the formidably defended double membrane envelop of Gram-negative bacteria, when molecular hits emerging from screens using recombinant proteins as targets were tested on the whole bacteria at the hit-to-lead stage. While some chemical characteristics have emerged that aided passage across the Gram-negative envelop [16], chemically transforming these hits into cell-penetrating leads overwhelmingly failed. This is why few large companies remaining in the field are now trying to take the target-biased phenotypic screen route, whereby genetic constructs are used to report on the function of specific bacterial pathways. This approach is, however, still largely unproven at an industrial scale. It also requires skilled, well-trained molecular microbiologists and a significant and diversified early research investment to validate and translate many recent academic discoveries on important, essential bacterial pathways into effective screening approaches. Despite its likely appropriateness, this approach is therefore akin to creating a scientific translation infrastructure where individual SMEs can rarely afford to explore, validate and screen for novel antibiotics discovery in even one pathway.

5.1.2. Access to New Natural Products/Chemical Libraries/Chemical Spaces, Including Virtual Ones, Are Needed

In view of their diminishing returns since the 1980s, every major pharmaceutical company has terminated their labour-intensive natural product efforts, such as most recently Novartis with the closure of its antibacterial department in 2018. While most of these strain and extract libraries still exist, they are now often, at least in part, publicly funded and hence under-equipped and under-staffed for the industrial-scale efforts required. To avoid screening the same chemical space over and over again, which effectively terminated the golden age of antibiotic discovery, extract dereplication is needed, but this is time-consuming and requires costly fractionation and analysis. Academic research budgets often cannot cover or justify these costs. Furthermore, the expansion of these libraries is also urgently required to maximise their novel discovery potential. However, this is also well beyond the reach of individual SMEs and this may well be the reason why many of them have recently turned instead to unconventional approaches including biological methods. Combining in silico compound libraries with artificial intelligence has recently

proven helpful in discovering new antibacterial compounds [17], suggesting this approach should also be further explored.

*5.2. At the Development Level*

(a) The cost of defining new target product profiles (TPPs) and keeping them relevant as ABR progresses;

(b) Associated costs in developing new types of preclinical and clinical testing procedures while drugs with novel types of activities are being developed;

(c) The cost of discussing and often co-developing new regulatory guidelines and trials with regulators where required;

(d) The cost of the commercialisation of new drugs classes working with novel mechanisms clinicians and healthcare professionals may initially be unfamiliar with and possibly find less trustworthy.

## 6. Economic Challenges of Developing and Commercialising Novel Antibacterial Compounds

The development and commercialisation of new antimicrobials is challenging at every level (Figure 2). Out of 15 novel antibacterials in development, only one has an impact on patients. Furthermore, companies that developed five of the last 15 new antibiotics approved by the Food and Drug Administration (FDA) in the past decade have gone through bankruptcy or near-zero valuations in the past two years, making them particularly fragile from a financial point of view.

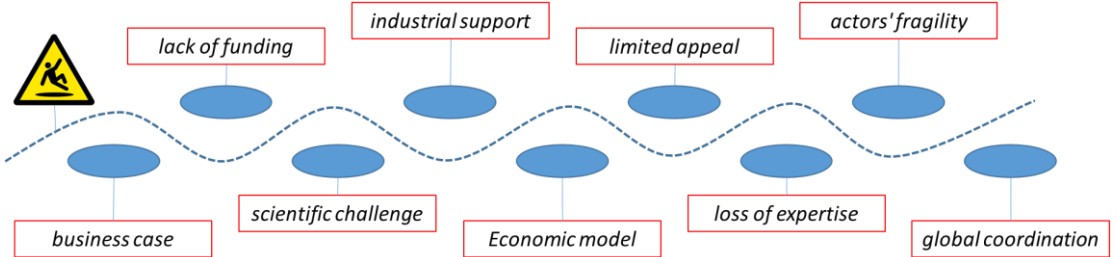

**Figure 2.** Challenges related to developing novel antibacterial compounds.

*6.1. There Is No Compelling Business Case for Developing Novel Antibacterials*

There is no compelling business case for developing novel antibacterials, and hence a business model is difficult to develop. The major challenge lies in poor economic incentives for investment within this area. In case of antibiotics, the sales-based model does not fit into a pure public health mandate to "steward" the availability of novel antibacterials and preserve their efficacy. Health authorities may want to keep a novel antibacterial in reserve, until other options are exhausted, to slow down the development of resistance against the novel antibacterial and secure it as a "last resort" option in case of an outbreak. This is what Outterson and Rex refers to as the "fire extinguisher" concept [18]: health authorities want to have novel antibiotics on a high shelf in each of their pharmacies in the hope they never have to use them. While the availability of such effective and cheap interventions is great for all of us, there is no economic incentive to develop new, better and potentially more expensive "extinguishers", because there would be no effective market for them.

*6.2. Traditional Drug Reimbursement Systems Are Not Effective for Antibiotics*

Commercially, the current drug reimbursement systems, which link the return on investment to the sales volume, are not effective R&D incentives for antibiotics, where low sales prices are the historical norm and for which a short duration of treatment and therefore low volume usage are generally desired. Pharmaceutical companies are reluctant to invest in the development of drugs that either do not generate enough profit or cannot be

sold rapidly. This is because they need to maintain a constant source of income to preserve value on the global stock markets and hence continue to be able to raise funds for their development of drugs in other traditionally or usually more lucrative areas such as chronic diseases, oncology and rare genetic disorders.

### 6.3. International Pharmaceutical Company Support for Antibiotic Research Has Dwindled

This is in part due to the limited profit made from the creation of novel antibiotics. The recent announcement of the withdrawal of Novartis from antibacterial development follows other large pharmaceutical companies like Allergan, AstraZeneca, BMS and Sanofi Aventis. This is in addition to the prior exit of companies such as Eli Lilly, which pioneered the manufacturing of the famous antibiotics penicillin, vancomycin and erythromycin. Currently, only GSK, Merck, J&J, Roche and Pfizer remain as pharmaceutical companies still operating R&D programs in antibacterial compounds.

### 6.4. The Majority of Antibacterial Compounds in Preclinical Trails Are Now Developed by Vulnerable SMEs

In addition, some 60% of these SMEs have no available product for the market, meaning they have no external income stream. Therefore, these companies have limited available investment, and the slightest change in the outcome of a trial can lead to significant loses and even closure [19].

### 6.5. There Is An Intrinsic Lack of Funding of Late-Stage Research for Novel Antimicrobial R&D

It is important to differentiate pre-phase-1 (Ph1) and post-Ph1 (i.e., late stage) funding availability. In the last five years, organisations like The Wellcome Trust, The Bill & Melinda Gates Foundation, The Combating Antibiotic Resistant Bacteria Biopharmaceutical Accelerator (CARB-X) and the Novo Repair (Replenishing and Enabling the Pipeline for Anti-Infective Resistance) Impact Fund have begun offering "push funding" to enhance development, but only for the pre-Ph1 stage up to and including Ph1 R&D stage. In practice, scientifically solid preclinical stage projects now get funded within one year by any one of the funders previously mentioned, although the coordination of submission and review procedures amongst them would certainly be a more effective approach. However, the significant challenge now lies with the Ph1 stage and beyond. Several alliances were formed recently to fund the development of new antimicrobials, such as the Biomedical Advanced Research and Development Authority (BARDA) Biopharmaceutical Accelerator (US Government), the Global Innovation Fund (Governments of the United Kingdom and China, the Bill and Melinda Gates Foundation) and the Research and Development of Global Antibiotics Partnership (GARDP). However, none of these funders can aspire to cover the cost of developing and commercializing multiple new drugs beyond the Ph1 stage in parallel. However, this is exactly what the world currently requires. Indeed, the costs significantly increase at this stage, 10 to 50 fold from the Ph1 to Ph3 to Ph4 stage and eventually to market. The development cost of novel antimicrobials was estimated at USD 1.5 billion [20]. This is mainly due to the cost of large, end-stage clinical trials, where Ph3 trials account for more than 80% of the total resources. Therefore, while several SMEs have made promising early-stage progress recently, where their next stage funding will come from is now not immediately clear.

### 6.6. SMEs Still Involved in Antibiotic Development Are Becoming Unviable

Those SMEs that in spite of the difficult economic conditions managed to raise venture capital to bring their portfolio compounds to market, including several even having closed their research departments to concentrate on development, still have to declare bankruptcy. In 2019, Achaogen, Aradigm, Melinta Therapeutics and Tetraphase Pharmaceuticals all stopped trading, and then their assets, including their drug IP rights, were sold as a fraction of their real value. Achaogen took 15 years and spent approximately one billion dollars gaining the FDA approval of Zemdri (plazomicin), a new antibiotic to treat complicated urinary tract infections. Zemdri was so promising that it became the first antibiotic, which

the FDA designated as a breakthrough therapy, expediting the approval process. The company also had other promising antibiotics in its portfolio; however, with limited profits, Achaogen filed for bankruptcy in April 2019. Only a few months later, the WHO added Zemdri to its list of new essential drugs. Achaogen is not the only example. In 2019, Aradigm failed to raise enough funds to complete Ph3 clinical trials for its inhaled ciprofloxacin to treat severe lung infections. Melinta, a company founded by prominent scientists including a Nobel Prize winner—Thomas Steitz, having Baxdela (delafloxacin) approved in 2017, two antibiotics developed in Ph3 trials and one developed in Ph2 trials, stopped operating in December 2019. None of these companies could realize the investments they made initially, as their new "fire extinguisher" drugs just did not sell [21]. They entered the novel antibacterial race, optimistically hoping the antibiotic market would change over time. However, a dysfunctional system remained in place and ultimately brought to the end of their drug development programs.

### 6.7. "Pull" Incentives Have Been Too Slow to Materialize

The conclusion is, therefore, that market forces combined with national and international regulations have actively inhibited the development and commercialization of novel antibiotics. To counteract this, in 2012, the U.S. representatives voted in favor of the Generate Antibiotic Incentives Now (GAIN) Act, as part of the Affordable Care Act. The GAIN act provides several competitive advantages to companies who bring an antibiotic to market. These advantages include fast track approvals and an exclusivity window to ensure substantial revenues and appropriate antibiotic use. However, in the last few years, these mechanisms were not developed further in the U.S. or elsewhere [22]. Indeed, the Developing an Innovative Strategy for Antimicrobial Resistant Microorganisms (DISARM) Act recently failed in the U.S. Congress again. By allowing higher Medicare (i.e., insurance-derived) reimbursement for new antibiotics, requiring hospitals are required to receive increased payments to monitor the use of the drugs and report data to the U.S. Centers for Disease Control and Prevention. The legislation has the potential to stabilize the antibiotics market, through spurring the development of new drugs and preserving the effectiveness of existing ones. To remedy the situation greatly and thus make it necessarily largely public purse-derived, "pull" incentives, otherwise known as "market entry rewards" (MERs), were lobbied by stakeholders. However, they have failed to materialize globally, despite a boarder awareness and understanding of the ABR crisis by political decision-makers. Only in the UK and Sweden, so far, has development of "delinking strategies" occurred, such as that of an "antibiotic susceptibility bonus" (ASB) [23–28]. Indeed, while governments and health services lean towards the prudent use of antibiotics, pharmaceutical companies need to increase sales volumes to maximize profits. An ASB seeks to realign pharmaceutical companies with antibiotic conservation by making a staged bonus a MER for antibiotic developers when resistance to their drug remains low over time. This type of bonus could address the lack of stewardship focus in a standard MER. At the end of 2020, the UK brought into practice the first-ever pilot scheme for an antibiotic subscription model. Its health agencies identified two new antibiotics, Cefiderocol (Fetcroja from Shionogi) and ceftazidime with avibactam (Zavicefta, from Pfizer), as suitable for purchase with a delinked model, in which the payment to the innovator of up to 1.7 billion USD is a fixed annual, estimated by the "absolutely required costs" of any new antibiotic from discovery to 10 years on market, by which time its sales might be expect to increase as ABR does, and made by any type of company in high-income countries [29]. However, the UK government could cap the reimbursement fee up to 10 M GBP/drug/year. Therefore, many more countries will need to implement this sort of scheme to support the market. In the U.S., the DISARM Act and more recently the PASTEUR Act (September 2020), which offer a predictable path to provide funding between 750 million to 3 billion USD for the creation of an impactful new antibiotic, are still navigating their way through the U.S. political systems. Moreover, pull incentives need to be on the agenda of the G7 or G20 countries as the evidence of their commitment.

*6.8. There Is a Limited Amount of Direct Antibacterial Research Currently Ongoing in Academia*

After the initial academic discoveries by Domagk of sulfonamides first by Fleming, then by the "Oxford Group" (Florey, Chain, Abrahams, Newton and Hodgkin) of penicillins and cephalosporins and those by Schatz and Waksman of streptomycins, the large-scale antibacterial drug discovery process that gave rise to the golden age of antibiotics became exclusive to pharmaceutical companies. Today, there are only a few academics left in this area, although they often publish high-profile work, for example chemists Gary Challis (Warwick) and Andy Myers (Harvard), the chemical biologists couple Dan Kahne & Susan Walker (Harvard) and the microbiologists Gerry Wright & Eric Brown (MacMaster), Kim Lewis (Northwestern) and Jeff Errington (Newcastle). Some have past industrial experience, and most have initiated or own SMEs of their own. Unfortunately, there is often limited contact and limited funding available for direct collaborations between them and other SMEs or large pharmaceutical companies still in the field.

*6.9. There Is an Increasing Loss of Expertise in Antibiotic Discovery*

As pharmaceutical companies previously dedicated to ABR research closed their antibacterial departments, the scientific experts involved in the development of many of our well-known antimicrobials have retired, leaving a knowledge gap to the next generation. Unfortunately, younger scientists who join SMEs also experience such a dilemma, for example Cubist Pharmaceuticals who has produced a number of powerful new antibiotics, such as Daptomycin. This is indirectly (for instance, through the lack of industrial endowments) influencing the interest of universities in supporting these areas of research. Together, these facts limit the preservation and transfer of fundamental knowledge, skills and experience combining microbiology with drug discovery. Fortunately, the Wellcome Trust in recognition of this has recently launched a call for a "landscape review" of this field in order to help design and fund ways to enable researchers in academia and SMEs to access trainings on relevant skills, access relevant expertise and resources/infrastructure at different stages of product development and raise funding for R&D programmes, with the aim of strengthening the antibiotic drug discovery pipeline [30].

*6.10. Meeting the Challenging, Often Internationally Diverse Regulatory Approval Requirements Is Especially Arduous and Demands Significant Investment, Time and Resources*

The practical consequences of meeting the regulatory approval requirements are beginning to have a negative impact. New drugs like omadacycline and plazomicin are not registered in Europe, and lefamulin still awaits a commercial partner to launch in Europe. For lefamulin and plazomicin, a key driver appears to be the fact that the cost of EU-required pediatric trials has exceeded any plausible estimate of sales in Europe, leading the companies to withdraw from the process. These events suggest that it is likely that new antibiotics from small companies will not be immediately available in Europe [31].

*6.11. There Is Limited Global Coordination towards Antibiotic Discovery*

When faced with a global emergency, such as COVID-19, the necessary remedy becomes necessary in global scientific, clinical and industrial collaboration for most stakeholders [32]. The development of novel antimicrobials for human, animal, agriculture, aqua culture, food and environmental use is also urgent and therefore requires global coordination, which is still limited. Moreover, there is no globally accepted harmonized mandate to support antibacterial R&D, unlike what was developed more than 15 years ago for vaccines [33]. Even though different approaches to stimulating antimicrobial R&D are being applied, those are neither recognized nor shared across stakeholders in different countries. It is important to inform health authorities across the world of the importance of this coordination and the importance of their role. Likewise, R&D efforts on novel antibacterial drugs (and also on antimicrobial diagnostics and vaccines) are not shared equally across the world. The lack of a united political framework is a noteworthy deficiency, especially in developing and underdeveloped countries.

## 7. The Meaning of "Market Failure"

Even a company that successfully commercializes a new, innovative antibacterial with significant potential is unlikely to survive in the current reimbursement climate. Indeed, having just one or a few first-class antibiotics in Ph3 trials or beyond as the main resource in their portfolios now evidently marks these companies out for a predetermined demise. As a consequence, future investment in this area is increasingly difficult to obtain, and existing SMEs in the area are choosing to increase their chances of survival by decreasing the proportion of their resources dedicated to infectious diseases research. This will only continue to propagate, as innovative small companies with promising new antibiotics continue to struggle and ultimately fail to bring products to market. This is the very embodiment of the economic term "market failure"—a market that perversely and pervasively discourages innovation when there is a clear global need.

## 8. Antibacterial Compounds Are a Global Common Good

According to the Stanford Encyclopaedia of Philosophy, in political discourse, the "common good" refers to those facilities, whether material, cultural or institutional, that the members of a community provide to all members in order to fulfil a relational obligation they all have to care for certain interests they have in common. Examples in modern society are shown as following: road systems; public parks; police protection and public safety; judicial systems; public schools; museums and cultural institutions; public transportation; civil liberties; the system of property; clean air and water; national defence and, of course, healthcare systems. The term itself can refer either to interests that members have in common or to facilities that serve common interests. Therefore, evidently, just like current novel antivirals and vaccines against COVID-19, to avoid an ABR "pandemic", i.e., the global spread of extremely drug-resistant (XDR) or PDR bacterial pathogens, working antibiotics are a global "common good". As such, innovative means of financing and making resources available at national and international levels have to be immediately considered.

## 9. The Need for Coordinated Global Approaches to Improve Antibacterial R&D

Global coordination is required to bring together aligned groups, such as SMEs, academic, clinical institutions and NGOs within the antimicrobial development landscape. Since 2015, the tripartite union of WHO/OIE/FAO launched the Global Plan of Action on Antimicrobial Resistance intending to address the problem of ABR. According to this plan, each member country must develop a strategic plan to address ABR following five objectives, including increasing investment in new medicines, vaccines and diagnostic tools. The pharmaceutical industry recently founded the AMR Industry Alliance, a coalition to provide biotechnological and R&D solutions to AMR problems. In Europe, the BEAM Alliance regroups and represents about 70 SMEs developing novel antibacterials. In 2017, the United Nations established an Inter-Agency Coordination Group on Antimicrobial Resistance [34]. The group's purpose is "to ensure sustained effective global action to address antimicrobial resistance, including options to improve coordination". The Global AMR R&D HUB was launched in May 2018 to enhance the international coordination of AMR research and development. It recently released another global platform in the pharmaceutical industry: a dynamic dashboard that continuously collects and presents information on more than 7 billion USD AMR R&D investments in bacterial infections of humans and animals. It also presents information on antibacterial products in the clinical development pipeline and incentives for antibiotic R&D. It has thus become the global knowledge center for ABR R&D, enabling for the first time ever any party or state to perform free, high-quality evidence-based analysis and decision-making to enhance global collaboration and coordination in this area. However, to increase the pace of antibacterial R&D in practice, a proactive, multifaceted approach must now be considered. Several proposals have recently been made [35], which are described as follows.

### 9.1. Towards More Pull Incentives?

Immediate and short-term strengthening of pull incentives [36], and in some cases salvage, promotes the delivery of existing opportunities in the current pipeline. However, Outterson and Rex recently recognized that the political will to develop MERs is difficult to muster, because large pharmaceutical companies would likely be major, and in some cases considered inappropriate beneficiaries (given recent high-profile cases of exorbitant medicine prices [37]) of, taxpayer funds. Alternative solutions have been considered, including: (a) traditional for-profit companies; (b) nonprofit enterprises; and (c) public benefit corporations, akin to public utilities. It was highlighted that nonprofit and public benefit corporations accept low-profit expectations, a key element of the antibiotic market at least in the short-to-medium term. They are also amenable to public–private partnerships, perhaps even externally overseen. Indeed, the latter two might crucially be more politically attractive recipients of pull incentives [38].

### 9.2. An Executional Global AMR R&D Hub?

Adams and co-workers further suggested the concurrent medium-to-long-term establishment of a global antibiotic research and development institute. A more practical version of this idea, due to its rapidly assembled, flexible and less costly advantages, is a virtual "accelerator", as was initiated recently by leading global medical charities for COVID-19 therapeutics and vaccines. The Global AMR R&D Hub with a representative global membership of global stakeholders could become a world leader in antibacterial R&D execution and drug discovery in addition to acting as an international think-tank. Every country in the world could contribute relative resources based on their per capita income, which in turn would guarantee prompt access to new drugs. Oversight and governance would need to be inclusive and transparent, including involvement from pubic, charitable and private organisations. Specific membership could include charities, public funders, academics, public or private natural product libraries, pharmaceutical companies, SMEs, contract research organisations (CROs), clinical trial locations and medical experts, IP specialists and even regulator envoys or their representatives. Its main task would be to select, prioritise and coordinate a single global pipeline delivering first-in-class antibiotics targeting key human health gaps, from discovery up to market entry.

### 9.3. A Global Antibacterial Compound R&D Advisory Group?

This could be formed from a small number of permanent Hub staff, each representing a type of stakeholder, although independent from undue influence and lobbying.

### 9.4. New, More Flexible Reimbursement Mechanisms?

Returns on investment would be agreed in advance with each member and crucially in the concept stage of any project well before successful arrival to market. There should be a transparent decision-making process, guaranteed by an independent monitoring body and authority to make and maintain legal agreements and associated frameworks.

Overall, this should facilitate interaction, synergy and active collaboration between academic, clinical and private research teams, minimising replication of facilities and effort and sharing obtained and existing knowledge ultimately providing an efficient framework to minimise negative impacts such as delays due to lengthy funding searches and associated financial difficulties.

If a global infrastructure as outlined begins to deliver, the next steps could be as follows.

### 9.5. Towards More Streamlined, Coordinated and Focused Clinical Trials?

The way we govern clinical trials needs to be globally connected and ultimately streamlined to help deliver effective and efficient studies. In the last few years, the FDA has reorganised its approach towards antibiotic development programs [39]. However, this needs global adoption, such as coordination and collaboration with the European Medicine Agency and the Japanese and other worldwide equivalents in order to maximise

efficiencies [40]. A significant focus should be on how clinical development, trial programs and approval procedures can be redesigned, streamlined and coordinated with global CDCs to target the constantly evolving landscape caused by MDR pathogens.

*9.6. Support for the Antimicrobial Valuechain in Manufacturing, Sales and Beyond?*

High-resource countries have already started to support development and supply of low-demand, novel antibiotics. Sweden recently entered into agreements with four companies, to guarantee through specified warehousing to ensure Swedish healthcare has effective access to selected antibiotic products. In return, companies will receive a guaranteed annual income per product [41]. Five antibiotics are included in this pilot scheme: Zerbaxa (ceftolozan-tazobactam), Recarbrio (imipenem-cilastatin-relebactam), Fetcroja (cefiderocol), Vaborem (meropenem-vaborbactam) and Fosfomycin infectopharm (fosfomycin). France and Germany have in place schemes that allow antibiotics that can demonstrate not "therapeutic superiority" (nearly impossible given the efficacy of historic drugs) but "added benefits" (such as a new mode of action that bypasses existing resistances) to negotiate higher prices with their national health systems [42]. However, as ABR continues, such national schemes are likely to increase access inequalities, rather than remedy a global problem. Therefore, we should consider setting up one global, governing body for manufacturing and "sales" of novel antimicrobials, this would include the stakeholders previously discussed in addition to those who use antibacterials within the one-world health landscape. New antimicrobials registered after successful clinical trials should be supported by appropriately resourced countries in order to meet the consumer needs of those membership countries.

## 10. Perspectives beyond Antibiotherapy

Ultimately, antibiotic therapy alone is unlikely to resolve the entire resistance problem. This is why alternative approaches are emerging, among which we can mention the inhibition of virulence factors, combination therapies based on collateral sensitivity, immunomodulation and antiresistance vaccination or phage therapy.

## 11. Conclusions

The consequences of increasing globalisation are the increasing dissemination of pre-existing health problems, such as emerging viral infections and ABR. However, as the COVID-19 pandemic has demonstrated, international political, scientific and industrial collaboration and global solidarity, humility and generosity are our most effective, in fact probably our only viable "exit strategies". An effective example is the AMR Action Fund which was announced in June 2020 [12]. It was created by leading pharmaceutical companies and is operationally supported by the International Federation of Pharmaceutical Manufacturers and Associations (IFPMA). Initially, the AMR Action Fund "will invest more than 1 billion USD it has already raised in smaller biotech companies and provide industry expertise to support the clinical development of novel antibiotics". However, beyond this, it "aims to bring 2–4 new antibiotics to patients by 2030". To achieve this, it will have to raise considerably more funds and therefore seeks "to bring together a broad alliance of industry and non-industry stakeholders, including philanthropies, development banks, and multilaterals, to encourage governments to create market conditions that will enable sustainable investment in the pipeline", which in practice remains entirely to be defined. However, we note with optimism that, given the alignment of stated missions, this could be the ideal funding instrument for the aforementioned Executional Global AMR R&D Hub.

**Funding:** This research received no external funding.

**Acknowledgments:** AJB would like to thank Eric Bacqué, Stéphane Renard, Lloyd Payne and Karen Lackey of Evotec for extensive discussions of many issues outlined herein in the last two years.

**Conflicts of Interest:** The authors declare no conflict of interest.

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
