# Peer review of "Looking for Solutions to the Pitfalls of Developing Novel Antibacterials in an Economically Challenging System"

_2036-7481, doi:10.3390/microbiolres12010013_

Round 1
Reviewer 1 Report
The article i received for review is interestingly written and presents very important information and point of view on the growing problem of the lack of research into new antibacterial substances.
After reading this manuscript, I have a number of comments that I would like to make to the Authors of the manuscript and the editorial Staff.
The CDC abbreviation appears on page 2. It appears only once in the article, so I propose to use the full name.
The abbreviation ABR appears for the 29th time in the text. I suggest giving the full name from time to time. For example, in the chapter title, instead of "Five core strategies to combat ABR", write "Five core strategies to combat rising antibacterial resistance".
On page 3, the author gives five core strategies: so instead of a) b) c) ... I propose 1) 2) 3) ...
On page 5, the author presents a very interesting graph of new antibiotics (Fig. 1). At the same time I write above that the golden age (it should be the "golden age" or the golden decade looking at the time span). However, I am wondering why the author, referring to Fig. 1, gives the range of the years 1950-1960. According to the information read from the graphs, the range of the years of the "golden age" should be 1940 - 1970.
The acronym FDA appears on page 9. It should be clarified the first time (Food and Drug Administration). Additionally, the author sometimes reports to the FDA and sometimes to F.D.A. I propose to standardize the abbreviation.
On page 17 there is the sentence "... collects and presents information on more than 7 bn USD ..." If the abbreviation bn means billions then I suggest using the full name, not the abbreviation.
Nowadays, in which the majority of the public and the scientific community is focused on the fight against COVID-19, a voice reminding us about the other threat lurking on us from multi-antibiotic-resistant bacteria is a voice needed.
Reviewer 2 Report
Brief Summary
The authors examined the underlying economic and administrative hurdles of the slow development of novel antimicrobials and proposed several possible initiatives to promote the economic incentives and feasibility of novel antimicrobial development.
Broad Comment
- The articles provided a good review of the economic and administrative hurdles in the development of novel antimicrobials. Some of the points stated are insightful and can provide scientists a good understanding in the economic and administrative difficulties in the development of novel antimicrobials.
- The article was written in a very layman language. In addition, there were a number of ambiguity due to the incorrect use of English. The authors may consider sending the article to professional scientific editors for improving the readability of the paper.
- It will be good if the authors can specifically state in the Teaser the arguments that they would like to content. This will be easier for readers to examine whether the arguments are valid logically and is well supported with evidences.
Specific Comment
As no line number is indicated, original sentences are copied in below to mark my comments.
- "the companies running these HTS screens, which were translated initially from successful in vitro drug screens on eukaryotic targets, chose to use
their small molecule libraries in bacteria as well. " --> It is unclear about the meaning of "HTS screens were translated initially from successful in vitro drug screens on eukaryotic targets". What does the term "translated" referring to? - "This is why the larger few companies remaining in the field now are now trying to take the biased phenotypic screen route". --> the authors are suggested to explain briefly about biased phenotypic screen route so that it will be easier for readers to understand the hurdles behind this approach.
- There is no caption / elaboration for Figure 2. Some of the term such as "actor fragility" may be unfamiliar to scientists.
- "Therefore, many more countries will need to implement this sort of scheme to rescue the market!" --> the use of exclamation mark is inconsistent with the objective and neutral nature of academic article.
- "It is important to educate health authorities across the world on the importance of this coordination and the role they must play in it" --> consider to change the verb "educate" to "inform"
Reviewer 3 Report
The article makes a detailed analysis of the economical problems that are impairing the development of novel antibiotics and presents the authors' opinions on how these problems can be faced and potential solutions for fostering again antibiotics' discovery. Reviews than include opinions are more relevant than purely reviewing articles, because the first offer ideas to explore forward, beyond what is already known. In this regard, this is a relevant paper that can have an important impact in future developments in the field. While all the economic aspects of the problem are well developed, some aspects dealing with novel approaches (not just antibiotics) as antivirulence, combination based in collateral sensitivity or anti-resistance vaccination among others, are less discussed and, including a section on these appraches to tackle antibiotic resistance will make a stronger review.
